# JOINTLY OPTIMIZING WIRELENGTH AND THERMAL FIELDS FOR CHIP PLACEMENT

## ABSTRACT

Macro placement is a crucial and complex issue in chip design. In recent studies, reinforcement learning (RL) has demonstrated outstanding performance in optimizing chip wirelength, but this leads to thermally inefficient design. Additionally, due to the specialized expertise necessary for creating chip benchmarks and the constraints imposed by confidentiality agreements, there exists a scarcity of publicly available chip thermal placement benchmarks. This work introduces a reinforcement learning-based thermal placement model that can optimize both wirelength and max temperatures. We also strictly followed the chip design process and established a macro thermal placement benchmark. This significantly reduces the entry barriers for researchers, facilitating benchmarking and result replication. Compared to other models, our model notably diminishes the chip's max temperature of the chip while slightly extending wirelength on smaller-scale chips. On large-scale chips, our model can further reduce wirelength while decreasing the chip's max temperature. Our code and benchmarks will be open sourced soon.

## 1 INTRODUCTION

With the development of large-scale integrated circuits (IC), placement is a crucial task that directly affects chip performance, such as speed and energy costGarg & Shukla (2016). In the placement task, macros (more than 100) and standard cells (more than 10k) are placed in appropriate locations to meet the design metrics such as wirelength, routability, timing, power, max temperature, and manufacturabilityQiu et al. (2023). As chip sizes continue to increase, manual design struggles to meet various design metrics simultaneously. Therefore, finding an automatic and efficient chip placement method becomes crucial.

Recent chip placement approachesChen et al. (2008); Lu et al. (2014); Lin et al. (2019); Cheng et al. (2018); Viswanathan et al. (2007) have shown significant advantages in wirelength optimization. However, As shown in Figure 1(d), shorter wirelength often results in an aggregation of components that increases max temperatures of chip then impacting chip performance. Max temperatures and temperature gradients have a definite effect on the reliability and performance of integrated circuits. For example, large temperature gradients increase clock skew in clock distribution networksXia et al. (2017), and device overheating due thermal runaway can occur in semiconductor devices due to the positive feedback between high temperature and increasing leakage currentMolter et al. (2023); Li et al. (2005).

Recent methods rarely prioritize heat optimization as a primary objective in chip placement. Existing approaches that simultaneously optimize wirelength and max temperature mainly focus on chiplet placement, and relatively few research focuses on macro placement. These works primarily suffer from the following three shortcomings: **First, Two-Stage Optimization Leads to Suboptimal.** As showen in Figure1(a) ,almost all thermal placement methods Ma et al. (2021); Chiou et al. (2023) divide the optimization process into two steps: first optimizing wirelength, and then optimizing max temperatures. This approach significantly restricts the model's exploration space during heat optimization, as component placements are nearly fixed, leading to local optima. **Second, These heat optimization methods are less effective for scenarios with high-density components.** Thermal placement methods typically use techniques like translation, rotation, and swapping after placing all components on the chip canvas to reduce max temperature. However, as shown in Figure 1(b),

when component density is too high, it greatly limits the space available for translation and rotation, significantly impacting optimization results. **Third, there is a lack of macro thermal placement benchmarks.** Thermal placement benchmarks are primarily focused on chiplet placement, with relatively fewer benchmarks addressing macro thermal placement. Generating these benchmarks requires running Electronic Design Automation (EDA) flows, which necessitate expertise in chip design—making data generation costly Jiang et al. (2024). Additionally, non-disclosure agreements (NDAs) for manufacturing techniques and EDA tools limit the release of raw data. As a result, most studies are only able to create small internal datasets or set the power density to a random value between $10^5$ and $10^7$ $W/m^2$ for technology validation, thus making benchmarks and reproducing results highly challenging.

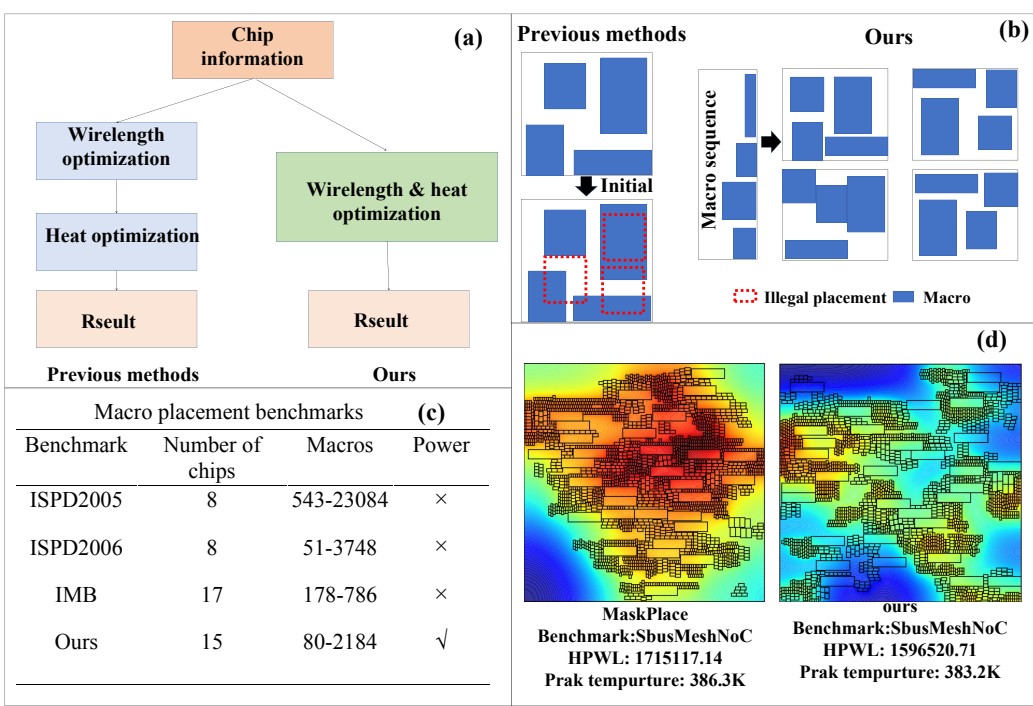

Figure 1: **Figure (a)**, wirelength and heat optimization process. Previous methods refers to Ma et al. (2021); Chiou et al. (2023). Our method combines wirelength optimization and heat optimization into one step, optimizing thermal performance from the beginning of the placement. **Figure (b)**, heat optimization strategies. Previous methods have a small feasible solution space when components are densely, and cannot significantly change the placement of components. Our model can generate a variety of placements during the heat optimization process. **Table(c)**, Commonly used macro placement benchmark. Commonly used macro placement benchmarks lack power information for each component. We built a macro thermal placement benchmark that includes detailed power information for each component through logic synthesis. **Figure(d)**, placement result visualization. Our model significantly reduces the max temperature and HPWL in large-scale chips.

To address these issues, we first developed a macro thermal placement model and then established an open-source benchmark for macro thermal placement. We utilize a reinforcement learning-based approach to simultaneously optimize wirelength and max temperature. By processing the wire-maskLai et al. (2022) and the max temperature under the current placement through a convolutional neural network, we select the most optimal placement positions for overall performance, minimizing both wirelength and max temperature, rather than optimizing them separately. In constructing the benchmark, we strictly followed the standard chip design process, performing logic synthesis on 15 real open-source chips to obtain detailed component information, rather than assigning random power values to each macro. Compared to existing benchmarks, our benchmark increases the theo-

retical power consumption pe macro while preserving the gate-level netlist through logic synthesis. The main contributions of this paper are as follows:

- We developed a reinforcement learning model that optimizes max temperature and wirelength simultaneously, taking into account constraints on both wirelength and max temperature to achieve a global optimal solution for wirelength and max temperature. This approach avoids the local optima caused by separately optimizing wirelength and max temperature.

- Through a comprehensive EDA process, we created the first open-source macro thermal placement benchmark, which provides a reliable baseline for comparison and replication, significantly lowering the barrier to entry for chip thermal placement.

- On the 15 benchmarks we have established, our model achieves the lowest placement temperature, demonstrating the effectiveness of our model.

## 2 RELATED WORK

### 2.1 CHIP PLACEMENT METHODS

There are two main optimization indicators commonly used in chip placement: using wirelength as an optimization metricChen et al. (2008); Lu et al. (2014); Lin et al. (2019); Cheng et al. (2018); Viswanathan et al. (2007); Kim et al. (2012); Kim & Markov (2012); Spindler et al. (2008); Chan et al. (2006); Lai et al. (2022); Shi et al. (2024); Mirhoseini et al. (2021) and optimizing both max temperature and wirelength simultaneouslyMa et al. (2021); Chiou et al. (2023).

**Using wirelength as an optimization metric.** The optimization of wirelength is primarily categorized into classic methods (e.g., analytical methods)Chen et al. (2008); Lu et al. (2014); Lin et al. (2019); Cheng et al. (2018); Viswanathan et al. (2007); Kim et al. (2012); Kim & Markov (2012); Spindler et al. (2008); Chan et al. (2006); Shi et al. (2024) and learning-based methods (e.g.,RL)Lai et al. (2022); Mirhoseini et al. (2021). Those methods typically utilize min $Wirelength(s, H)$ as the objective function, with some incorporating additional objectives such as congestion and overlap. For example, DREAMPlaceLin et al. (2019) utilizes analytical methods to optimize wirelength and density, convert the placement task into min $WA(s, H) + \lambda Density(s, H)$. $WA$ denotes the smoothed weighted average wirelength used to approximate Half Perimeter Wire Length (HPWL), $Density$ denotes the differentiable density measure used to penalize overlap, and $\lambda$ is the trade-off factor. The problem is then solved numerically using classical mathematical optimization techniques, such as gradient descent, to rapidly generate a high-quality complete placement. MaskPlaceLai et al. (2022) utilizes reinforcement learning to optimize wirelength. Reinforcement learning views the placement process as a Markov Decision Process (MDP). In each step $t$, a component is placed on the chip canvas. Set the reward as $r_t = HPWL_{t-1} - HPWL_t$ and train the model to maximize the reward in order to achieve the min $Wirelength$. Those methods can achieve placement results that surpass humans in terms of wirelength optimization, but they overlook the impact of max temperatures on the chip, leading to a reduction in the chip's thermal performance.

**Optimizing both thermal and wirelength.** Some methods consider the chip's thermal distribution during placement and optimize wirelength and max temperature simultaneously. However, these methods primarily concentrate on 2.5D chiplet placement and relatively less on macro placement. TAP-2.5DMa et al. (2021) employs simulated annealing to discover a placement result with improved wirelength and max temperature by moving the chiplet through translation and rotation from the initial layout generated byChen & Chang (2006). SA cost function is $cost = \alpha T + (1 - \alpha)W$,where T and W are temperature and wirelength respectively. $\alpha$ is the balance coefficient. $\alpha = 0$ when $T \leq 85$. Then, $\alpha$ incrementally rises with T until it peaks at a maximum value of 0.9. This means that heat optimization is withheld during wirelength optimization until the temperature surpasses 85 degrees Celsius, at which point heat optimization is initiated. Chiou et al. (2023) utilizes an SP-based tree to achieve wirelength-focused placement. After the placement is completed, perform post-placement with thermal considerations. However, these approaches prioritize wirelength optimization initially and subsequently address heat optimization once wirelength optimization reaches a certain level. If the interplay between the wirelength and thermal parameters is not considered initially, the system will converge to a locally optimal solution. Furthermore,

compared to macro placement, chip placement requires fewer components (about 10), yet the larger number of macros (more than 100) notably expands the solution space.

## 2.2 CHIP DATASET AND BENCHMARK

Datasets and benchmarks are crucial for the development of research. This facilitates benchmarking and result reproducibility, while also reducing the barriers to entry for new researchers. Researchers have developed various datasets tailored for specific tasks to foster advancements in chip design. In the prediction task, CircuitNet and CircuitNet 2.0Jiang et al. (2024); Chai et al. (2023) collected over 10,000 data points from CPU, GPU, and AI chips, and conducted multi-model prediction tasks such as timing, routing feasibility, and IR-drop prediction. In macro placement tasks, generally used public benchmarks include ISPDNam et al. (2005; 2006), IBM benchmarkAlpert (1998), and the Ariane RISC-V CPU designZaruba & Benini (2019). These benchmarks mainly include the length and width and pin position (the components are interconnected through pins.) of each component, and the topological relationships (Netlist) of components. Due to confidentiality reasons, those benchmarks do not disclose specific power of individual components. Due to the absence of specific power for each component, thermal field for macro placement cannot be conducted. While the power density of each macro can be constrained within the range of $10^5 - 10^7 W/m^2$ based on statistical principlesCong et al. (2004), the power density of components varies across different manufacturing processes and component librariesBorkar (1999); Wrzecionko et al. (2009); Ku et al. (2007); Hanson et al. (2003); Li et al. (2005); Kim et al. (2005). Additionally, the power of each macro is also influenced by voltage fluctuations and frequencies. In other words, the power of macros is influenced by various factors and requires precise calculations to obtain more accurate results.

## 3 PRELIMINARY AND NOTATION

### 3.1 MACRO PLACEMENT

Macro placement is an integral part of placement. The macro placement task can be viewed as an optimization problem. The objective function is minimized by adjusting the position of the macro while satisfying certain constraints. In macro placement, common optimization metrics mainly include wirelength and max temperature, aiming for the chip to have the shortest possible wirelength and the lowest temperature. Consider with the challenges associated with directly calculating wirelengths, recent work primarily relies on Half Perimeter WireLength (HPWL) to approximate wirelength which is computed by accumulating all the half-perimeters of bounding rectangle of all the nets from the chip netlist. As the chip power is predominantly generated in macros, we calculate the power density of each macro based on its power and area. Subsequently, we obtain the chip's thermal field through finite element analysis, extracting the max temperature from it.

Placement constraints mainly include: **overlap**, which avoid overlapping between each macro, In the chip canvas, each position can be occupied by at most one macro. **congestion**, the congestion of each position's routing in the chip canvas should be less than a fixed threshold. Therefore, the entire placement optimization problem can be formulated as:

$$\min_{x,y} HPWL(x,y) + \alpha MaxT(x,y) \tag{1}$$

$$s.t. Overlap(x,y,w,h) = 0 \tag{2}$$

$$Congestion(x,y,w,h) \leq C \tag{3}$$

$HPWL(.)$ means Half Perimeter WireLength, and $MaxT(.)$ represents the highest temperature of the entire chip, $Overlap(.)$ and $Congestion(.)$ represent the methods for calculating overlap and congestion, respectively. $(x,y) = (x_1, y_1, x_2, y_2...x_n, y_n)$ represent the placement position of $i^{th}$ macro.

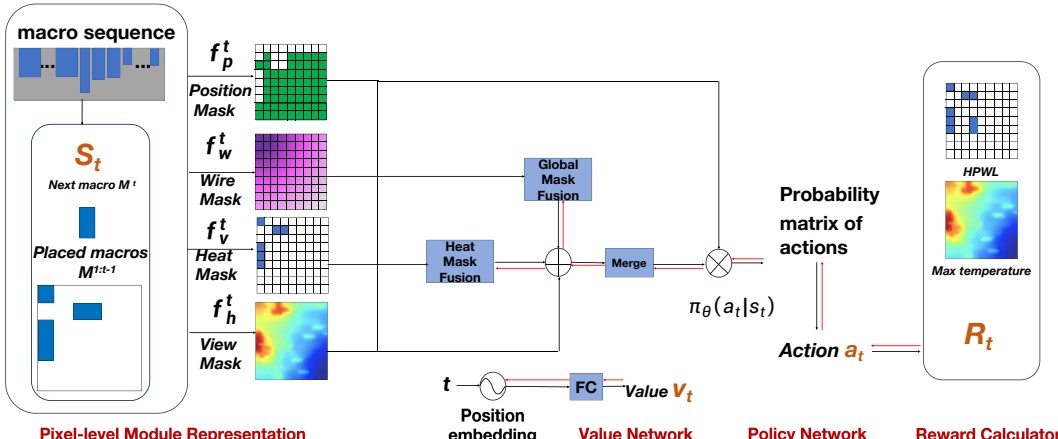

Figure 2: The overall structure of our model. Black arrows represent the forward propagation process, while red arrows represent the backward propagation process.

# 4 THERMALLY DRIVEN MACRO PLACEMENT MODEL

By placing a macro into the chip canvas each step, we transform chip placement into a Markov decision process (MDP)Kaelbling et al. (1996). The overall architecture of the model, as shown in Figure2, consists of a policy network $\pi_\theta(a_t|s_t)$ and a value network $V_\phi(s_t)$. The policy network adopts an encoder-decoder structure, using the previous state $s_t$ as input to select an action $a_t$ as output. Black arrows represent the forward propagation process, while red arrows represent the backward propagation process. The reward calculator computes the reward by weighting the increments of post-placement HPWL and max temperature.

## 4.1 HEAT MASK

In this section, we complete thermal simulation of the entire chip through finite element analysis (FEA) methods. In the FEA process, accurately describing the boundary effects of a heat source on thermal load variations is crucial. Geometric mapping refers to the formulation of how to construct mechanical analysis models from level-set-based geometric structuresChen et al. (2023); Guo et al. (2014); Kang & Wang (2013); Zhang et al. (2015); Kreisselmeier & Steinhauser (1980); Wang et al. (2018); Torii et al. (2022). We use density-based mapping to maintain a certain level of accuracy while considering the avoidance of additional costs and extra implementation work associated with grid re-partitioning. We use the efficeint algorithm based on Green functionLiu et al. (2013) to map 3d thermal field to 2d. The Heat equation can be expressed as:

$$\sigma \frac{\partial T(r,t)}{\partial t} = \nabla \cdot (\kappa \nabla T(r,t)) + p(r,t) r \in D \tag{4}$$

In our model, we assumes all four sides of the chip are insulated from the ambient environment. The heat flow towards to x- and y- direction walls is zero. Heat generated from components on chip can be dissipated toward to heat sinks at the top or PCB at the bottom. The boundary condition of our system can be expressed as follows:

$$\frac{\partial T(r,t)}{\partial x}\Big|_{x=0,L_x} = \frac{\partial T(r,t)}{\partial y}\Big|_{y=0,L_y} = 0 \tag{5}$$

$$\kappa \frac{\partial T(r,t)}{\partial z}\Big|_{z=-L_z} = h_p T(x,y,-L_z,t) \tag{6}$$

$$\kappa \frac{\partial T(r,t)}{\partial z}\Big|_{z=0} = -h_s T(x,y,0,t) \tag{7}$$

$h_p$ denotes primary heat flow to the heat sink and $h_s$ denotes secondary heat flow to the PCB respectively. In our model, the chip is divided into M by N bins. More bins imply more detailed temperature distributions. Accord to the Laplace's equation of heat equation, the general solution without time with respect to the boundary condition in equation 5, 6, 7 has been derived then implemented in integral function to approximated the temperature distribution of bins in chips. Solution in z-direction described in is independent to solution in x- and y-directions. Dimensionality reduction in the z-axis directions significantly faciliates heat analysis on temperature distribution on chip.

We introduce 3d finite element analysis (FEA) method to analysize the thermal distribution on chips. A level set function (LSF) $\phi(x)$ is introduced to describe the shape of components. In our chip designs, each macro is approximated as rectangle. Which LSF can be constructed in unified form as:

$$\phi(x, y; x_0, y_0) = 1 - (\frac{x - x_0}{a})^m - (\frac{y - y_0}{b})^m \tag{8}$$

Where m is integer number which controls the components shape; a and b are semi-major length and semi-minor length of component respectively; $(x_0, y_0)$ corresponds to the geometric center coordinate of component. To account for a unified FEA process without remeshing grids after each movement of macros, the geometric description function of our macros is projected onto a density field with the Heaviside function:

$$H(x) = \begin{cases} 1, x > 0 \\ 0, x \leq 0 \end{cases} \tag{9}$$

The region that Heaviside function equal 1 represents the occupancy of components, where the heat source load is distributed. The heat source intensity function (HSIF)$\Phi(x)$ in whole design can be expressed as:

$$\Phi(x) = \sum_{c=1}^{N_c} Q_c(x) \cdot H(\phi_c(x)) \tag{10}$$

Where $Q_c(x)$ is the intensity distribution function of the ith heat source. The structured quadrilateral finite elements are introduced in our FEA design. The element equilibrium equation is

$$\mathbf{K}^e \mathbf{T}^e = \mathbf{P}^e \tag{11}$$

Where $\mathbf{K}^e$ is the element heat transfer matrix, is the elemental nodal $\mathbf{T}^e$ temperature vector, $\mathbf{P}^e$ is the equivalent elemental nodal thermal load vector, respectively. The entire chip's thermal field is obtained through finite element analysis, and this thermal field is used as a heat mask input for the model.

## 4.2 REINFORCEMENT LEARNING

We drew inspiration from the network structure of MaskPlaceLai et al. (2022) and used the Heat Mask along with the Position Mask, Wire Mask, and View Mask as inputs to the network. We utilized the commonly used PPOSchulman et al. (2017) framework to train the policy $\pi_\theta(a_t|s_t)$. We combine the HPWL and the weighted max temperature of the entire chip as the reward. Specifically, we use the increase in wirelength and temperature after placing each component as negative rewards to minimize wirelength and highest temperature. The reward calculation method is as follows:

$$r_t = (HPWL_{t-1} - HPWL_t) + \alpha(Tmax_{t-1} - Tmax_t) \tag{12}$$

$HPWL_{t-1}, HPWL_t, Tmax_{t-1}, Tmax_t$ represent the wirelength and highest temperature at time t-1 and time t, respectively. $\alpha$ is a hyperparameter used to balance the magnitudes of the two parameters.

## 5 THERMALLY DRIVEN MACRO PLACEMENT BENCHMARK

Chip design is a complicated process, primarily divided into two stages: front-end design and back-end design. In front-end chip design, the main focus is on describing the functionality of the chip by using hardware description codes like Verilog to illustrate its logical functions. Recently, some work has used large language models (LLM) to generate Verilog codeLai et al. (2024); Alsaqer et al. (2024); Chang et al. (2023), significantly accelerating the front-end design process of chip design. Following that, the Verilog codes link to top module hierarchically in design are analyzed then mapped to gate-level descriptions through logic synthesis with respective to specified design constraints. After logic synthesis, the connection relationships between macros and standard cells are established. There are differences in the area and timing parameters of standard cells and macros under different technology library. After logic synthesis, the chip design progresses into the back-end design stage, where the focus is primarily on completing the physical design of the chip. This includes tasks such as floorplanning, placement and routing. Placement and routing are key steps in chip back-end design. During this stage, optimizing the performance, power, and area (PPA) of the chip through placement and routing optimizations is crucial, much of the research has been focused on fundamental trade-offs made in semiconductor design for PPA. In this section, we construct

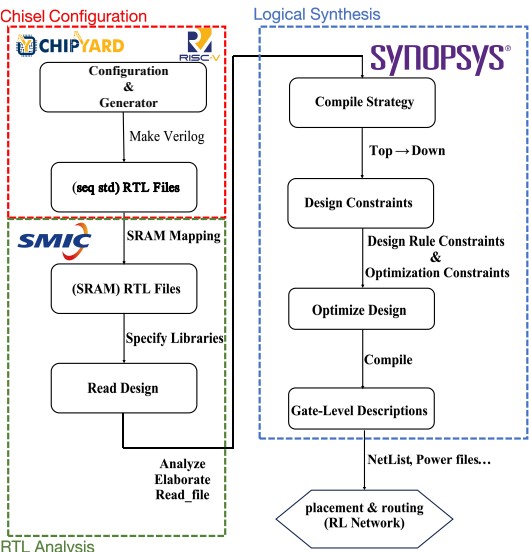

Figure 3: Generation process of the thermally driven macro placement benchmark.

benchmarks for chip thermal placement as shown in Figure 3. We use RISC-V SoC RTL design tools in chipyardAmid et al. (2020), an opensource framework for SoC agile development. All the designs are the variants of RISC-V SoC with the core (s) being RocketAsanovic et al. (2016) or (and) BoomCelio et al. (2015) as well as Shuttle. The benchmark designs are generated from chipyard implemented with Verilog HDL. We apply SRAM compiler to map the cache modules consist with sequential cells in Verilog files to vendor SRAMs. The Verilog files with SRAM modules are logical synthesized using Synopsys Design Compiler to get the gate-level netlists as well as the power and area of components. The SMIC 55-nm technology node is adopted to memory compiler and standard cells during the logical synthesis progress of our research. We obtain 15 benchmarks in total and the detailed information of each benchmark is listed in Table 1.

The netlist generated by logical synthesis is used to represent the logical relationship among components in integrated circuits. The number of pins for each cell is determined by the finout and finin of cells or macros. We distribute the pins location on macro boundary randomly. A netlist in a design can be defined as $H(V, E)$, in which $V$ represents to the vertices of components in hypergraph $H$. The nets correspondes to the hyperedges E. The power data generated by logical synthesis is used to represent the heat power of cells and macros. The heat power for each component consists with static and dynamic power. We introduce the dynamical power in macro placement tasks since the dynamic power is orders of magnitude larger than the static power of macros.

Table 1: The detailed information about the benchmark.

| Benchmark | Macros | Std cells | Nets |
|---|---|---|---|
| Rocket | 80 | 203034 | 302279 |
| HwachaRocket | 162 | 809553 | 1090468 |
| Sha3Rocket | 80 | 230981 | 353109 |
| LargeBoomAndRocket | 138 | 1191052 | 1581156 |
| SmallBoomAndRocket | 90 | 571904 | 800541 |
| DualBoomAndDualRocket | 260 | 2287113 | 3006245 |
| DualBoomAndRocket | 180 | 2169563 | 2836397 |
| GemminiRocket | 392 | 1145387 | 1700468 |
| MempressRocket | 824 | 697697 | 1133408 |
| FPGemminiRocket | 280 | 1262227 | 1752083 |
| GemminiShuttle | 289 | 1204401 | 1756288 |
| LeanGemminiRocket | 392 | 852394 | 1277971 |
| QuadRocketSbusRingNoC | 552 | 888764 | 1327155 |
| SbusMeshNoC | 2184 | 1978414 | 3229628 |
| SbusRingNoC | 936 | 1322864 | 1998056 |

## 6 EXPERIMENTS

We test our method on benchmark in Table 1 and compared it with MaskplaceLai et al. (2022). We set trade-off coefficient of wirelength and max temperature as $\alpha$=1 and 0 respectively. The other hyperparameters set same with previous work. We notice that the number of macros in our benchmarks vary from 80 to 2184. For the SbusMeshNoC benchmark which has over 2000 macros and over 3000000 nets, a single RL epoch by step-by-step placement costs more than an hour. Thus, for large benchmarks (contains over 300 macros) we select 256 macros in train process then generate all macros finally.

**Main results.** Table 2 gives the detailed results of each method with same benckmark. The max temperature is lowest in our method with $\alpha$=1 for most of benchmark. The exception in SbusRing-NoC and MempressRocket might be associated with the train process in which we only select 256 macros for large scale chips. Among these benchmarks, HwachaRocket performs best heat optimization which reduce the max temperature 25.76K. Other benchmark also perform well in heat optimization task compared with maskplace. The optimization of max temperature of chip is attributed to the heat reward in equation 12.

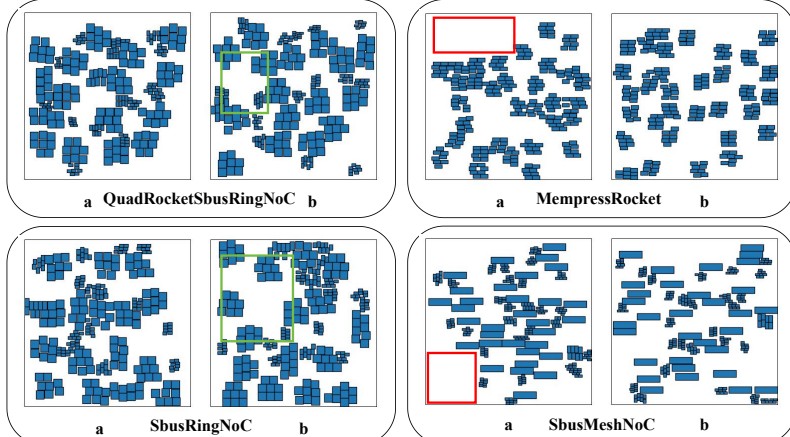

Figure 4: Results of the placement for the first 256 macro. a represents the results without heat optimization, while b represents the results with heat optimization. The red box represents the space not utilized by the model without heat optimization, while the green box indicates that our model has reserved more ample space for the placement of subsequent macros.

Table 2: Comparison of HPWL ($10^5$) and max temperature.

| Benchmark | Methods | HPWL ($10^5$) | Max temperature (K) |
|---|---|---|---|
| Sha3Rocket | maskplace | 5.90 | 393.57 |
| | ours ($\alpha$=0) | **4.97** | 397.32 |
| | ours ($\alpha$=1) | 5.95 | **388.93** |
| Rocket | maskplace | 5.73 | 399.84 |
| | ours ($\alpha$=0) | **5.09** | 395.79 |
| | ours ($\alpha$=1) | 5.58 | **392.06** |
| SmallBoomAndRocket | maskplace | 5.56 | 422.55 |
| | ours ($\alpha$=0) | **5.28** | 423.54 |
| | ours ($\alpha$=1) | 6.97 | **413.91** |
| LargeBoomAndRocket | maskplace | 1.17 | 446.23 |
| | ours ($\alpha$=0) | **1.06** | 443.60 |
| | ours ($\alpha$=1) | 1.50 | **435.83** |
| HwachaRocket | maskplace | 1.55 | 452.64 |
| | ours ($\alpha$=0) | **1.40** | 454.96 |
| | ours ($\alpha$=1) | 2.05 | **429.20** |
| DualBoomAndRocket | maskplace | 1.70 | 479.73 |
| | ours ($\alpha$=0) | **1.49** | 472.55 |
| | ours ($\alpha$=1) | 1.87 | **466.87** |
| DualBoomAndDualRocket | maskplace | 2.22 | 487.77 |
| | ours ($\alpha$=0) | **2.19** | 495.62 |
| | ours ($\alpha$=1) | 2.89 | **484.05** |
| FPGemminiRocket | maskplace | **3.92** | 348.68 |
| | ours ($\alpha$=0) | 4.04 | 349.00 |
| | ours ($\alpha$=1) | 4.40 | **347.37** |
| GemminiShuttle | maskplace | 8.21 | 354.01 |
| | ours ($\alpha$=0) | **7.81** | 356.69 |
| | ours ($\alpha$=1) | 8.19 | **345.93** |
| QuadRocketSbusRingNoC | maskplace | 24.68 | 342.45 |
| | ours ($\alpha$=0) | 23.02 | 342.17 |
| | ours ($\alpha$=1) | **22.62** | **341.28** |
| SbusMeshNoC | maskplace | **15.51** | 401.98 |
| | ours ($\alpha$=0) | 17.15 | 386.39 |
| | ours ($\alpha$=1)) | 15.97 | **383.26** |
| SbusRingNoC | maskplace | 27.04 | 364.77 |
| | ours ($\alpha$=0) | 26.53 | **359.68** |
| | ours ($\alpha$=1) | **25.91** | 360.75 |
| MempressRocket | maskplace | 132.53 | **324.24** |
| | ours ($\alpha$=0) | 142.83 | 328.94 |
| | ours ($\alpha$=1) | **128.93** | 328.42 |

For wirelength results, the most of our results shows longer wirelength indicates the balance between wirelength and thermal properties of chip. However, we notice as chip scale increases, the difference of wirelength between our method and other method decreases. For QuadRocketSbusRing-NoC, SbusRingNoC and MempressRocket. the wirelength for our method is lower than previous methods. Figure 1(d) shows the temperature distribution as well as placement result in SbusMesh-NoC benchmark, We observe that, due to the heat optimization in the reward function, macros are distributed more evenly across the canvas.

The results of placing the first 256 macros in our model are shown in Figure 4. We notice that compared to models without heat optimization, our model has a larger internal space and higher space utilization rate, which is more conducive to the placement of subsequent modules. Due to entropic order for macros introduced by "repulsive force" equivalent to heat optimization reward function in our methods, the space between large macros has large and uniform size, small macros within different modules can be placed in these free space effectively. This indicates that performing

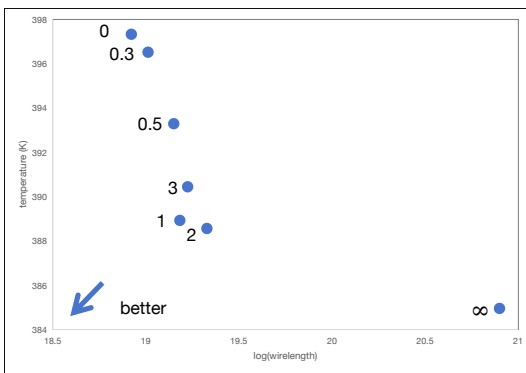

Figure 5: trade-off between wirelength and max temperature.

heat optimization at the initial stages in large-scale chip placement tasks has a positive impact on reducing max temperature and wirelength.

**Trade-off results.** In fact, the macro placement in benchmark is associated with trade-off coefficient $\alpha$ between wirelength and max temperature. Accord to scatter plot in Figure 5 As $\alpha$ increases, the temperature decreases significantly with wirelength increases slightly which can be attributed to the heat optimization for placement. However, as trade-off coefficient tends to 1, the result of wirelength and max temperature shows chaotic. The chaotic of trend might be associated with the expansion of configuration for macro in phase space. We should notice that lower wirelength indicates the aggregation between macros. Heat optimization tends to separate the macros in whole canvas to deminish hot spot. The configuration of macro placement increases massively indicates we need more epoch in train process to explore the phase space of macros. To avoid the chaotic introduced by expansion of configuration of macro, we select $\alpha$=1 as trade-off coefficient.

# 7 CONCLUSION

In this paper, we developed a reinforcement learning-based macro placement model that optimizes both wirelength and thermal field, thus achieving a balance between wirelength and max temperature. Furthermore, we established 15 open-source macro thermal placement benchmarks through a comprehensive EDA process. We obtained gate-level netlists and detailed power information for each macro through logic synthesis. From the experiments, it is evident that our model can reduce the chip's max temperature while slightly increasing the wirelength on smaller-scale chips. On larger-scale chips, our model disperses the initial component placement through heat rewards, providing ample space for subsequent macros and reducing wirelength to a certain extent. This also demonstrates the importance of early heat optimization in large-scale chip placement. We aim for our benchmark to promote research in chip thermal placement, thereby enabling chips to achieve optimal performance.

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
