# OpenReview forum: "Jointly Optimizing Wirelength and Thermal Fields for Chip Placement"
_ICLR.cc/2025/Conference — ICLR 2025 Conference Withdrawn Submission_

### Official Review · Reviewer_4wnr · 2024-10-26

**Soundness:** 2
**Presentation:** 1
**Contribution:** 2
**Rating:** 3
**Confidence:** 5

**Summary:**

The paper presents a reinforcement learning-based method for macro placement with a focus on thermal considerations and introduces a benchmark for evaluating power and thermal effects in macro placements. These contributions are aimed at fostering further research in thermal-driven placements. The effectiveness of the proposed HeatMask is demonstrated through experiments and ablation studies.

**Strengths:**

1. The introduction of the novel HeatMask, which effectively captures temperature increases to guide the placement algorithm, is a significant innovation.
2. The development of a comprehensive benchmark for power and heat evaluation is a valuable contribution to the EDA community.

**Weaknesses:**

1. The paper may not fully bridge the awareness gap for readers from the AI community regarding the interplay between thermal issues and power in chip placement. A more thorough discussion on how placement affects maximum chip temperature would be beneficial. Typically, thermal considerations have been less emphasized in 2D chip placement compared to 2.5D-3D stacking.
2. While thermal issues have been extensively studied in 2.5D chiplet and 3D IC designs, this paper could benefit from a more expansive review and comparison of thermal-aware placement techniques. Such a discussion in both the related work and experimental sections would underscore the significance and innovation of this study.
3. The technical novelty appears limited as the overall architecture closely mirrors that of the existing MaskPlace, with the primary innovation being the integration of HeatMask. Clarification of the unique contributions to the AI community would strengthen the paper.
4. The paper does not conclusively address whether optimizing macro wirelength and heat leads to enhanced overall chip performance. Since standard cell placements also play a crucial role in performance, focusing solely on macro placements may overlook broader performance impacts.
5. Established benchmarks like OpenROAD already support comprehensive RTL-GDS automation design flows, including power evaluations. The paper could enhance its impact by highlighting what new capabilities or improvements the proposed benchmark offers. Releasing the benchmark before submitting the paper might allow for a fairer assessment.
6. The experimental results do not consistently show superiority over the MaskPlace method. In some cases, the targeted heat optimization does not achieve better outcomes than the baseline.

**Questions:**

1. The manuscript would benefit from careful proofreading to correct various typos and formatting issues. The citation format, in particular, needs attention.
2. The label 'IMB' in Figure 1(c) appears to be a typographical error and might actually refer to 'IBM'.
3. The description on Page 1, Lines 37-39 seems to contradict the data shown in Figure 1(d), where shorter wirelength correlates with improved temperature performance.

---

### Official Review · Reviewer_uLf7 · 2024-10-28

**Soundness:** 2
**Presentation:** 2
**Contribution:** 2
**Rating:** 3
**Confidence:** 4

**Summary:**

This paper introduced an RL solution for thermal-aware chip macro placement.
Overall the paper is well written, with certain technique concerns detailed in the following review section.
The major contributions include:
1) RL framework considering chip thermal behavior.
and 2) thermal-aware benchmark suite for related research.

**Strengths:**

1. Thermal-aware macro placement using RL with explicit thermal reward calculation

2. Comprehensive experiments on multiple RISC-V chip designs.

**Weaknesses:**

1. I will challenge the technique novelty of this paper. As the paper only uses regular RL algorithms to solve an non-convex optimization problem. Adding additional terms in reward does not necessarily grant technical contributions. I could do more for the chip flow, like timing, and final PPA, etc.
2. The paper claimed both method and benchmark. However, none of them are discussed thoroughly in the manuscript. First the method of RL is nothing new. Second, authors did not provide more detailed analysis of the benchmark (How do they behave on various physical design solutions and how do they differs from existing benchmarks outside, as there are many designs come along with OpenROAD and ICCAD Contests.)
3. I am concerned about the the solution of the thermal-aware macro placement. it seems the reward calculation requires call of FEA every iteration, which is lacking in efficiency. Authors are recommended to include runtime comparison on each design. Also from the result table (Tab.2), we observe significant wirelength penalty on the proposed approach. 30% more HPWL on HwaChaRocket.
4. More baseline comparisons are necessary to demonstrate the effectiveness. For example, AutoDMP (https://d1qx31qr3h6wln.cloudfront.net/publications/AutoDMP.pdf), which also optimizes macro placement through a gradient way.

**Questions:**

1. In Table 2, are HPWL estimated only based on Macroplacement or after global placement? Can you also show the total wirelength as well as timing PPA, after the full flow? I assume the authors have already had access to synopsys tool.
2. How are the max temperature computed? There should be some power distribution on floorplans, which is missing in this paper.
3. As discussed in the weakness, please introduce a runtime breakdown for all the experiments. In general, physical design requires multiple runs to achieve desired results, the efficiency really matters.

---

### Official Review · Reviewer_aNur · 2024-11-01

**Soundness:** 2
**Presentation:** 2
**Contribution:** 2
**Rating:** 3
**Confidence:** 4

**Summary:**

This paper addresses the issue of macro placement in chip design, aiming to optimize both wirelength and thermal distribution. The authors argue that existing methods, which often focus primarily on wirelength optimization, can lead to high temperatures and thermal imbalances. They propose a reinforcement learning-based model that considers both wirelength and maximum temperature during the placement process. They also create a new open-source benchmark for macro thermal placement, which includes detailed power information for each component, enabling more realistic thermal simulations.

**Strengths:**

1. The paper introduces a new reinforcement learning model that optimizes both wirelength and thermal fields simultaneously, addressing a gap in current chip placement methodologies.
2. The authors develop the first open-source macro thermal placement benchmark, which is a valuable contribution to the field. This benchmark allows for standardized testing and comparison of different thermal placement methods.
3. The proposed model demonstrates a significant reduction in the maximum temperature of the chip compared to traditional wirelength-focused methods, especially for large-scale chips.

**Weaknesses:**

1. The authors may investigate the potential impact on other chip performance metrics like timing and area, other than wirelength.

2. The computational cost of the reinforcement learning model becomes significant for large-scale chips with many macros and nets. To address this, the authors train the model on a subset of macros for large benchmarks. However, this approach raises several questions:
* How is the subset of macros selected? Is it a random selection or based on specific criteria?
* Does training on a subset affect the model's ability to generalize and optimize the placement of all macros in the final design?

3. Limited Comparison with Existing Methods. The paper primarily compares the proposed model with MaskPlace, a wirelength-focused method. Including comparisons with other thermal optimization techniques would strengthen the evaluation. This may involve:
* Benchmarking against existing chiplet placement methods that consider thermal parameters.
* Adapting traditional macro placement algorithms to incorporate thermal awareness and comparing their performance.

**Questions:**

1. The description of the thermal simulation could be enhanced with clearer explanations of the finite element analysis (FEA) process, the dimensionality reduction technique, and the heat source intensity function (HSIF). It is especially critical for this paper since the authors claim to generate and release the dataset.

2. The paper states that the code and benchmarks will be open-sourced soon. It is crucial for the authors to provide timely access to these resources. Releasing the datasets will enhance the paper a lot!

3. Generalization to Other Technologies: The benchmarks are generated using the SMIC 55-nm technology node. Exploring the model's applicability to other technology nodes is important. This could involve:
* Generating benchmarks using different technology nodes and evaluating the model's performance.
* Analyzing the sensitivity of the model to technology-specific parameters like thermal conductivity and heat capacity.

4. Please fix the format of citations in Section 1.

---

### Official Review · Reviewer_2x5d · 2024-11-02

**Soundness:** 2
**Presentation:** 2
**Contribution:** 2
**Rating:** 3
**Confidence:** 2

**Summary:**

Applications of RL to chip placement with joint optimization of wirelength and peak temperatures. PPO is used where the policy networks guide the macro placement.  Use of a 3rd party FEA thermal solver to aid the learning process. Authors also mention the lack of open-source datasets, so they created their own. Improvement in max temperature relative to an existing established flow.

**Strengths:**

Interesting application of RL to an important problem in digital design and potentially valuable contributions to the ML community including the dataset itself if provided. PPO seems to be a good candidate for this problem and nice to see the usage of industry standard tools in the process like the FEA thermal solver.

**Weaknesses:**

Limited details on the learning aspect for the RL approach -- namely just Figure 2 and a brief discussion on reward function. The justification for the reward function is also a bit lacking w/o much experimental data.  I do not see any ablation studies or discussion of hyper parameter derivations as well.

The methods for the RL training including hardware profiles, simulation time, etc are missing.

Missing Very little effort in ensuring reproducibility / no supplementary material (they do mention will provide soon in abstract)

**Questions:**

1) Can you elaborate on why you chose PPO? Did you experiment w/ other algorithms?
2) Did you have to modify / play around w/ certain parts of the network architecture and PPO implementation to get your results?
3) Furthermore, can you provide justification for your network architecture and overall RL approach, perhaps in the form of ablation studies?
4) When do you plan to open source the code and benchmarks?

---

### Official Review · Reviewer_7mWC · 2024-11-04

**Soundness:** 2
**Presentation:** 1
**Contribution:** 2
**Rating:** 3
**Confidence:** 4

**Summary:**

This paper proposed a RL-based macro placement method that optimizes both HPWL and thermal. Specifically, it introduces an additional reward term that estimates the max temperature. It also generates a benchmark dataset for thermal-aware macro palcement.

**Strengths:**

1. This paper considers the thermal issue in macro placement, and proposes a RL-based method for placement.
1. This paper provides a new thermal placement benchmark.

**Weaknesses:**

1. The introduced method includes the emperature as an additional reward term. However, the temperature computation is based on a basic method and is not novel. Therefore, the technical contribution is minor.
1. It seems that standard cells are not considered in the placement process.
1. Previous methods are mainly designed for 2.5D or 3D chip design. This work, however, only considers 2D placement. Generally speaking, thermal management seems not a critical issue in 2D chip placement.
1. Previous thermal-aware methods, such as TAP-2.5D, are not compared. The only considered baseline, MaskPlace, is not designed for temperature optimization. Therefore, the comperasion is insufficient.
1. Many typos. For example, "Prak" should be "Peak" in Figure 1, "pe" in Line 108-109, $T^e$ in Line 306. Moreover, the citation is not in a proper form throughout this paper.

**Questions:**

1. Some existing benchmarks can be used for thermal placement, e.g., [ICCAD 2022 contest](https://www.iccad-contest.org/2022/) (3D Placement with D2D Vertical Connections) and the recent [ChipBench](https://arxiv.org/pdf/2407.15026). Why not using or comparing with these benchmarks?
1. In Table 2, "ours(\alpha=0)" achieves a slightly better HPWL than MaskPlace. Which component in the model contributes to HPWL improvement?

---

### Note · Authors · 2024-11-13

I have read and agree with the venue's withdrawal policy on behalf of myself and my co-authors.